# Natural Ligand-Mimetic and Nonmimetic Inhibitors of the Ceramide Transport Protein CERT

**DOI:** 10.3390/ijms23042098

**Published:** 2022-02-14

**Authors:** Kentaro Hanada, Shota Sakai, Keigo Kumagai

**Affiliations:** 1Department of Quality Assurance, Radiation Safety and Information Management, National Institute of Infectious Diseases, Shinjuku-ku, Tokyo 162-8640, Japan; 2Department of Biochemistry and Cell Biology, National Institute of Infectious Diseases, Shinjuku-ku, Tokyo 162-8640, Japan; sakais@nih.go.jp (S.S.); kuma@nih.go.jp (K.K.)

**Keywords:** intracellular transport of lipids, lipid transfer proteins, membrane contact, sphingolipids, Golgi apparatus, *CERT1*, *COL4A3BP*, START domain, structure-based drug design, off-target effects

## Abstract

Lipid transfer proteins (LTPs) are recognized as key players in the inter-organelle trafficking of lipids and are rapidly gaining attention as a novel molecular target for medicinal products. In mammalian cells, ceramide is newly synthesized in the endoplasmic reticulum (ER) and converted to sphingomyelin in the *trans*-Golgi regions. The ceramide transport protein CERT, a typical LTP, mediates the ER-to-Golgi transport of ceramide at an ER-distal Golgi membrane contact zone. About 20 years ago, a potent inhibitor of CERT, named (1*R*,3*S*)-HPA-12, was found by coincidence among ceramide analogs. Since then, various ceramide-resembling compounds have been found to act as CERT inhibitors. Nevertheless, the inevitable issue remains that natural ligand-mimetic compounds might directly bind both to the desired target and to various undesired targets that share the same natural ligand. To resolve this issue, a ceramide-unrelated compound named E16A, or (1*S*,2*R*)-HPCB-5, that potently inhibits the function of CERT has recently been developed, employing a series of in silico docking simulations, efficient chemical synthesis, quantitative affinity analysis, protein–ligand co-crystallography, and various in vivo assays. (1*R*,3*S*)-HPA-12 and E16A together provide a robust tool to discriminate on-target effects on CERT from off-target effects. This short review article will describe the history of the development of (1*R*,3*S*)-HPA-12 and E16A, summarize other CERT inhibitors, and discuss their possible applications.

## 1. Introduction

Lipids are the major constituents of all cell membranes and play dynamic roles in organelle structure and function. In eukaryotic cells, the endoplasmic reticulum (ER) is the main center for the synthesis of diverse types of lipids. Lipids newly synthesized in the ER are rapidly and accurately delivered to other organelles by a variety of lipid transfer proteins (LTPs), in a nonvesicular manner, at zones where the ER is in contact with other specific organelles [1,2,3,4,5,6]. Various viruses and obligate intracellular bacteria hijack LTPs of host cells for their proliferation [7,8,9,10]. Therefore, LTPs are fundamental for the metabolism of lipids and biogenesis of membrane compartments in cells and have been gaining attention as a novel type of molecular medicine target. Nonetheless, the repertoire of specific LTP inhibitors to date is limited.

Sphingolipids are a class of lipids that contain the long-chain amino alcohols termed sphingoid bases, including sphingosine, dihydrosphingosine, and phytosphingosine, as their structural backbone (Figure 1A). Sphingolipids are ubiquitous constituents of cell membranes in eukaryotes and also present in a limited number of prokaryote species [11,12,13,14]. Sphingolipids participate in various biological events, including cell growth, apoptosis, differentiation, and adhesion (reviewed in [15,16]). The choline-containing sphingophospholipid sphingomyelin is a ubiquitous and predominant type of sphingolipid found in mammals and is thought to be a chemically robust but noncovalently interactive type of phospholipid, unlike other types of phospholipid [11].

In mammalian cells, sphingomyelin is mainly located in the plasma membrane (PM), although the Golgi apparatus (where sphingomyelin is synthesized) and endosomes/lysosomes (to which the PM-derived endocytosis is directed) also have sphingomyelin [17,18]. In the PM phospholipid bilayer, sphingomyelin is predominantly distributed in the exoplasmic leaflet, with less found in the cytoplasmic leaflet. Cytoplasm-oriented sphingomyelin was recently shown to be pivotal for forming a PM-derived, concave, tubular structure [19]. Sphingomyelin, in concert with cholesterol, generates nano-scale membrane domains (so-called “lipid rafts” or “detergent-resistant membranes”), in which the spatiotemporal confinement of various molecules needed for signaling and other dynamic events of membranes can efficiently occur (see [20,21] for recent reviews). Sphingomyelin may interact with specific membrane proteins as their functional modulator, while sphingomyelin is also an important metabolic reservoir for sphingolipid mediators such as ceramide, sphingosine, and sphingosine-1-phosphate, which are produced as catabolites of sphingomyelin (reviewed in [15,22,23,24]).

The ceramide transport protein CERT, a typical LTP, mediates the transport of ceramide from the ER to the Golgi apparatus, where ceramide is converted to sphingomyelin [11,25]. CERT was first discovered in 2003. Although many LTPs capable of catalyzing the inter-membrane transfer of lipids in vitro have been identified, to the best of our knowledge, CERT is the first LTP that has been shown, with compelling biochemical and genetic evidence, to mediate the inter-organelle transport of membrane lipids. In 2001, a ceramide-mimetic compound that potently inhibits ER-to-Golgi transport of ceramide in cells was serendipitously developed [26]; it was later shown to be a selective inhibitor of CERT [27], as described in more detail below. This ceramide-mimetic inhibitor of CERT has been widely used as a pharmacological tool in laboratory-based studies. Nevertheless, there was inevitable concern that natural ligand-mimetic compounds may directly bind to the desired target but also to various undesired targets that share the same natural ligand. Hence, ceramide-nonmimetic inhibitors have recently been developed using a structure-based drug design approach [28]. The establishment of a set of ligand-mimetic and nonmimetic inhibitors has provided a pharmacological tool to discriminate on-target effects from off-target effects. In this short review article, we describe the CERT inhibitors for which activity has been validated within living cells; we also discuss possible medical applications of CERT inhibitors.

## 2. The Biosynthetic Pathway of Sphingomyelin in Mammalian Cells

Figure 1B summarizes the biosynthetic pathway of sphingolipids in mammalian cells (reviewed in [29,30,31]). L-serine and palmitoyl coenzyme A (CoA) are condensed to the labile intermediate 3-ketodihydrosphingosine by the enzyme serine palmitoyltransferase (Enzyme Commission number: EC 2.3.1.50), followed by its reduction by 3-ketodihydrosphingosine reductase (EC 1.1.1.102) to produce dihydrosphingosine. Dihydrosphingosine undergoes *N*-acylation by acyl-CoA:dihydrosphingosine *N*-acyltransferase, which is widely known as ceramide synthase (CERS) (EC 2.3.1.24), followed by Δ^4^-desaturation by the enzyme DEGS1 (EC 1.14.19.17) to generate ceramide. Of note, the *N*-acylation of dihydrosphingosine by CERS to produce dihydroceramide is predominant in the de novo pathway, whereas CERS also catalyzes the *N*-acylation of sphingosine (sphingosine is produced in the degradation of complex sphingolipids) to generate ceramide in the salvage pathway. The human genome encodes six isoforms of CERS (CERS1–6), and each CERS has a different acyl-CoA preference in the synthesis of ceramide [32]. In some tissues, such as the kidney and intestines, a substantial proportion of dihydroceramide is hydroxylated at the C4 position by the enzyme DEGS2 (EC 1.14.18.5), which dually catalyzes Δ^4^-desaturation and C4-monooxygenation of dihydroceramide to produce phytoceramide (whereas in plants, C4-hydroxylation of dihydrosphingosine occurs before its *N*-acylation). These reactions to produce ceramides are thought to occur at the cytosolic surface of the ER. Then, ceramide is delivered to the lumenal side of the Golgi apparatus and converted to sphingomyelin by phosphatidylcholine:ceramide cholinephosphotransferase (EC 2.7.8.27), which is widely known as sphingomyelin synthase (SMS). The human genome contains two isoforms of SMS, SMS1 and SMS2 (their official gene names in the human genome are *SGMS1* and *SGMS2*, respectively) [33,34]. SMS1, which plays the predominant role in the de novo synthesis of sphingomyelin, is localized to the distal Golgi regions (including medial-/trans-Golgi cisterna and possibly also the trans-Golgi network), while SMS2 resides in both the Golgi apparatus and the plasma membrane. SMS2 likely plays a crucial role in the re-synthesis of sphingomyelin from ceramide generated at the PM and also contributes to the de novo synthesis of sphingomyelin at the Golgi apparatus, redundantly with SMS1. Ceramide is also converted to glucosylceramide (GlcCer) by UDP-glucose:ceramide glucosyltransferase (UGCG) (EC 2.4.1.80). The GlcCer synthase UGCG is abundant in the proximal Golgi region, with cytosolic orientation of its catalytic site. After being transported to the lumenal side of the Golgi apparatus, GlcCer is further converted to more complex glycosphingolipids. The ER-to-Golgi trafficking of ceramide includes at least two pathways: CERT-dependent and -independent pathways. CERT mediates the ER-to-distal Golgi nonvesicular trafficking of ceramide, which is the major pathway for the synthesis of sphingomyelin but not GlcCer. CERT-independent pathways, including the one for GlcCer synthesis and the minor pathway for sphingomyelin synthesis, have not been well characterized. It also remains unclear how ceramide delivered by CERT to the cytosolic side of the Golgi membrane is relocated to the lumenal side.

## 3. Structure of CERT and Its Splicing Variants

CERT contains several functional domains and motifs (Figure 2) [25,27,35,36,37,38,39]: a pleckstrin homology (PH) domain, which preferentially binds phosphatidylinositol 4-monophosphate [PtdIns(4)P]; a serine-repeat motif (SRM); a “two phenylalanines in an acidic tract” (FFAT) motif, which binds vesicle-associated membrane protein-associated protein (VAP); and a “steroidogenic acute regulatory protein-related lipid transfer” (START) domain, which effectively encloses various natural ceramide types (i.e., ceramide, dihydroceramide, and phytoceramide) in its hydrophobic cavity at 1:1 stoichiometry for inter-membrane transfer. The CERT START domain may catalyze the inter-membrane transfer of diacylglycerol (DAG), but with less efficiency compared with ceramide [25,27].

When the SRM of CERT undergoes multiple phosphorylations, the function of CERT is repressed (Figure 2) [36]. Previous studies have shown that protein kinase D [41] and casein kinase 1γ isoforms [42] are involved in the phosphorylation of the SRM, while protein phosphatase 2Cε participates in the dephosphorylation of the SRM [43]. The phosphoregulatory aspect of CERT has recently attracted attention because specific missense mutations in the CERT-encoding gene, *CERT1* (also known as *COL4A3BP*, *CERT*, and *STARD11*), in the human genome, were found to be associated with intellectual disabilities and mental development disorders with dominant inheritance [44,45,46,47,48], and some of these mutations were demonstrated to compromise the functional repression of CERT [48,49].

Human *CERT1* produces at least three splicing variants of transcripts [48]. The *CERT1* transcript isoform encoding the 598-amino acid protein CERT, also known as Goodpasture antigen-binding protein (GPBP)Δ26, is ubiquitously expressed in human tissues. The isoform encoding a 624-amino acid protein, named CERT_L_ (also known as GPBP), which has an extra 26 amino acids just before the START domain of CERT (Figure 2), is also widely but less abundantly expressed in various tissues. CERT_L_, like CERT, mediates the ER-to-Golgi trafficking of ceramide in cells [25,48]. Both CERT and CERT_L_ form homo-oligomers in cells [40,50,51]. The third isoform is predicted to encode a 752-amino acid protein, with an extra fragment comprising 128 amino acids before the initial methionine of CERT_L_, but its transcription is absent or rare except in sperm cells [48]. The putative product encoded by this third transcript will not be considered further in this review article as it has not yet been characterized.

Revert et al. reported that CERT_L_, but not CERT, is exported from cells via a nonconventional secretory pathway [40,52]. Phylogenetic studies have suggested that clear paralogs of human CERT are present widely or ubiquitously in multicellular animals but are absent in other organisms [11,53]. Darris et al. proposed that CERT/GPBPΔ26 originated in the putative last common unicellular ancestor of metazoans, choanoflagellates, and filastereans during the evolution of life [53]. It should also be noted that, although both GPBP and GPBPΔ26 were originally reported to be atypical serine/threonine kinases [50,51], this assignment remains controversial because they do not have the signature sequence usually found in serine/threonine kinases [11,53].

## 4. (1. *R*,3*S*)-HPA-12, a Ceramide-Mimetic Inhibitor of CERT

The initial finding that a (3-hydroxy-1-hydroxymethyl-3-phenyl propyl)alkanamide (HPA) compound (Figure 3) was a ceramide-mimetic potent inhibitor of CERT was serendipity. In the late 1990s, SMS had not been purified nor had its cDNA been identified (cDNAs for SMS1 and SMS2 were identified in 2004 using genetic and bioinformatics approaches [33,34]). With the provision that if a specific inhibitor of sphingomyelin synthesis could be developed, it could be a tool to find the identity of a yet-to-be identified SMS, for example, by inhibitor-conjugated affinity purification of the enzyme, Hanada, Kobayashi, and their colleagues attempted to find SMS inhibitors among ceramide-mimetic compounds. Soon after beginning this attempt, a novel compound (named HPA-12, for an HPA that had the C_12_ alkyl chain) was found to potently inhibit the de novo synthesis of sphingomyelin, but not of other lipid types, in a radioactive serine-based metabolic labeling study of lipids in living cells [54]. Disappointingly, the compound did not inhibit SMS activity at all. Nonetheless, before starting the SMS inhibitor development project, another project being run by Hanada’s group had isolated an interesting mammalian cell mutant, named LY-A, in which de novo synthesis of sphingomyelin was compromised even when there was normal SMS enzyme activity [55] and revealed that LY-A cells are defective in ER-to-Golgi transport of ceramide [56]. The knowledge obtained during the characterization of LY-A cells naturally led the research group to the hypothesis that HPA-12 is an inhibitor of ER-to-Golgi transport of ceramide. This hypothesis was soon proved to be correct, based on several criteria used for the characterization of LY-A cells [54]. Specifically, HPA-12 impaired the ER-to-Golgi transport of C_5_-DMB-ceramide, a fluorescent analog of ceramide. When the ER and the Golgi apparatus were merged by treating cells with brefeldin A, HPA-12 no longer inhibited the synthesis of sphingomyelin. In addition, HPA-12 did not affect the ability of LY-A cells to produce sphingomyelin, which suggested that the molecular target of HPA-12 was the factor that was impaired in LY-A cells [25]. Thereafter, CERT was identified as the factor that was impaired in LY-A cells. Then, HPA-12 was biochemically demonstrated to be an antagonist of CERT [27], and co-crystals of the START domain (i.e., the ceramide-binding domain) of CERT in a complex not only with ceramide but also with HPA-12 were resolved [37,57].

HPA-12, which contains two chiral carbons, has four stereoisomers (Figure 3). It should be emphasized that there was a revision in the stereochemistry of active HPA-12. At the time when HPA-12 was initially developed, the absolute configuration of HPA-12 that inhibits ER-to-Golgi trafficking of ceramide was assigned as (1*R*,3*R*), according to the configuration of the material compounds and the supposed mechanisms of the reactions used for the chemical synthesis of HPA-12 stereoisomers [26]. However, it was later discovered that, unexpectedly, epimerization of the C3-position of HPA occurred during the synthesis reactions. In 2011, Ďuriš et al. used crystallization-induced asymmetric transformation technology to reveal that the absolute configuration of the HPA-12 stereoisomer that was initially assigned as (1*R*,3*R*) must be (1*R*,3*S*), [58], and this finding was verified by Kobayashi and his colleagues [59]. In addition, the deposited crystallographic data of the CERT START domain in complex with an HPA compound, which had been misassigned as the (1*R*,3*R*) type, was found to be more fit for the complex with its (1*R*,3*S*)-isomer [28,60]. It has now been established that the absolute configuration of HPA-12 that potently inhibits CERT is (1*R*,3*S*), and the compound previously identified as “(1*R*,3*R*)-HPA-12” has been relabeled as “(1*R*,3*S*)-HPA-12”.

## 5. Structure and Activity Relationship of HPA Compounds

Hereafter, to depict the absolute configuration of the potent CERT inhibitor HPA-12 as (1*R*,3*S*), we will rename the configuration of HPA compounds and their derivatives as necessary, although previous studies used the configuration assignments prior to their revision.

Among the four stereoisomers of HPA-12, only the (1*R*,3*S*) type displays a potent activity to inhibit de novo synthesis of sphingomyelin in cells. This is consistent with the affinity of these compounds for the CERT START domain. A surface plasmon resonance (SPR)-based assay determined the *K*_d_ values of the (1*R*,3*S*)-, (1*S*,3*R*)-, (1*S*,3*S*)-, and (1*R*,3*R*)-isomers to be ~30, >10,000, ~4500, and ~4000 nM, respectively (Figure 3) [28]. The C1 position of HPA-12 corresponds with the *N*-acyl chain-linking carbon in ceramide, and the 1*R* configuration in HPA matches the configuration of natural ceramide (Figure 3). This may explain why (1*R*,3*S*)-HPA-12 has a much higher affinity for the ceramide-binding START domain than its 1*S*-cognate, (1*S*,3*S*)-HPA-12. Not only the presence but also the configuration of the C3-position OH group is crucial for the activity of HPA-12 [54,61] (Figure 3). This feature can be accounted for by the prediction that the OH group at the C3 position of (1*R*,3*S*)-HPA compounds, but not its 3*R*-cognate, may form hydrogen bonds with Asn504 of the CERT START (Figure 4) [28], whereas Santos et al. have proposed that the OH group at the C3 position of (1*R*,3*R*)-HPA-12, as well as that of (1*R*,3*S*)-HPA-12, may form hydrogen bonds with Asn504 [62]. It should also be noted that adding one OH group at the C2 position of HPA-12 in either its 2*S* or 2*R* configuration reduces CERT inhibition activity [61], although natural ceramides (ceramide, dihydroceramide, and phytoceramide) are hydroxylated at the corresponding position (Figure 1 and Figure 3). This presumably reflects that Asn504 of the CERT START forms hydrogen bonds with the C2-position OH group of ceramides but with the C3-position OH group of (1*R*,3*S*)-HPA (Figure 4) [28,37].

The CERT inhibition activity of HPA series in living cells also depends on their *N*-acyl lengths. When acyl chains were examined from C_3_ to C_18_, carbon lengths of 11 to 15 were pivotal for strong activity (an acyl length of C_13_ was optimal), while lengths shorter or longer than the 11 to 15 carbon range resulted in a negative impact on activity [54,61]. The reduction in activity by shorter acyl chains is most likely due to fewer hydrophobic interactions of the compounds in the ceramide-binding pocket (Figure 3 and Figure 4) [37,57,63]. In contrast, the low activity of longer acyl chain HPAs is presumably due to their very poor miscibility with water and cell membrane permeability.

[^18^F]HPA-12, a radioactive fluorine-conjugated (1*R*,3*S*)-HPA-12 compound, was developed for use as an in vivo positron emission tomography (PET) probe of CERT in the brain [64]. Whereas [^18^F]HPA-12, like HPA-12, acts as a CERT inhibitor in living cells [64], it is unlikely to be useful as an in vivo probe for CERT because the biodistribution of [^18^F]HPA-12 in various regions of the (mouse) brain shows no close relationship with the distribution of CERT [65]. There is a possibility that [^18^F]HPA-12 may be useful as an in vivo probe to detect sphingolipid imbalance in the brains of patients with Alzheimer’s disease [65].

## 6. Other Ceramide-Mimetic Inhibitors of CERT

Using cell-free screening systems and/or in silico docking evaluations, several types of CERT inhibitors, with a ceramide-like core structure, have been developed [66,67,68], although it remains unclear whether they can serve as more potent CERT inhibitors than (1*R*,3*S*)-HPA-12 without affecting ceramide metabolism enzymes, including SMS, in cells. Interestingly, some existing medicines were recently found to act as CERT inhibitors that are predicted to fit the ceramide-binding pocket of the CERT START domain, although their structural resemblance to ceramide is marginal (Figure 5) [68]. Fluralaner is a systemic insecticide and acaricide, used in veterinary medicine, that acts via potent blockade of GABA- and L-glutamate-gated chloride channels (https://pubchem.ncbi.nlm.nih.gov/compound/Fluralaner; Accessed on 22 December 2021). Lomitapide is a cholesterol-lowering agent that acts by inhibiting microsomal triglyceride transfer in humans (https://pubchem.ncbi.nlm.nih.gov/compound/Lomitapide; Accessed on 22 December 2021). Considering that their activities to inhibit CERT are “off-target” effects from the perspective of their approved medical aims, these findings should provide invaluable mechanistic insights into previously enigmatic side effects of these drugs, but also suggest that “drug repositioning” (also referred to as drug repurposing and drug reprofiling), which is the scientific strategy of revisiting existing drugs and determining if they could be used for new therapeutic purposes, could be a feasible way to accelerate the application of CERT inhibitors for clinical use.

Following the pioneering work by Pagano and his colleagues [69], various types of fluorescent analogs of ceramide have been developed as fluorescence-microscopically visible probes that can, at least partly, mimic the behavior of natural ceramide in living cells (reviewed in [70]). Among them, C_6_-nitrobenzoxadiazole (NBD)-ceramide and C_5_-dimethyl BODIPY^TM^ (DMB)-ceramide (also known as BODIPY^TM^ FL-C_5_-ceramide) have been most widely used to date (Figure 5). Although both C_6_-NBD and C_5_-DMB types are recognized by CERT [27], only C_5_-DMB-ceramide displays a clear CERT-dependence for ER-to-Golgi transport in cells [25,56]. This is accounted for by the biophysics through which C_6_-NBD-ceramide (which is less hydrophobic than C_5_-DMB-ceramide) undergoes spontaneous inter-membrane transfer too rapidly to detect discernible CERT-dependent transfer [56,71]. Indeed, when molecular hydrophobicity is appropriately increased, an NBD-conjugated ceramide exhibits a CERT-dependent transport in cells and is metabolized to its sphingomyelin and GlcCer types although its metabolism is less efficient than C_6_-NBD-ceramide [63]. The above-described fluorescent analogs of ceramide have a hydroxyl group at the C1 position, which is the acceptor for the polar head groups to be converted to sphingomyelin and GlcCer. In contrast, 1-*O*-acetyl-C_16_-ceramide-NBD is not converted to sphingomyelin nor GlcCer, although it displays CERT-dependent transport in cells (Figure 5) [72], indicating that this compound is recognized by CERT. Although the primary purpose of fluorescent analogs of ceramide may be live imaging, one should be cautious in case they are also acting as “inhibitors of CERT” in cells by competing with endogenous, natural ceramide. Similar caution should be applied for other nonnatural ceramides, such as the short-chained C_4_- and C_6_-ceramides, which are recognized by CERT (Figure 3) [28].

The mycotoxin fumonisin B1, which has a sphingoid base-like structure, is a natural inhibitor of the ceramide synthases the ceramide synthases (Figure 5) [73]. A computational docking simulation study predicted that *N*-acylated forms of hydrolyzed fumonisin B1 (Figure 5) fit the ceramide-binding cavity of the CERT START domain, raising the possibility that a metabolite of fumonisin B1 may act as a ceramide-mimetic inhibitor of CERT [74].

## 7. E16A/(1*S*, 2*R*)-HPCB-5, a Ceramide-Nonmimetic Inhibitor of CERT

Ceramide is the common metabolic precursor for sphingomyelin and glycosphingolipids and the common product for their degradation pathways, meaning that various enzymes involved in sphingolipid metabolism recognize ceramide. In addition, ceramide acts as a directly binding modulator for various crucial factors, including protein kinase Cζ [75,76], kinase suppressor of Ras (KSR) [77], mixed lineage kinase 3 (MLK3) [78], protein phosphatase 2A (PP2A) [79,80], inhibitor 2 of PP2A (also known as SET and TAF-1β) [81], p53 [82], the porin-like mitochondrial protein VDAC2 [83], and the mitochondrial fission factor Mff [84]. Moreover, the pore-forming physiochemical nature of ceramides in the mitochondrial membrane has been proposed [85]. These facts have raised concerns that ceramide-mimetic CERT inhibitors such as HPA-12 may bind to CERT and various other undesired targets sharing the same natural ligand ceramide.

To improve this situation, CERT inhibitors with no apparent structural similarity to ceramides have recently been developed using a structure-based drug design strategy [28,86]. In the first step, a seed compound with no ceramide-like structure but with the ability to form a hydrogen-bonding network in the ceramide-binding START domain of CERT was obtained, following the virtual screening of approximately 3 × 10^6^ compounds. The seed compound was subjected to a series of in silico docking simulations, efficient chemical synthesis, affinity analysis with an SPR-based system, protein–ligand co-crystallography, and various in vivo assays. As a result of this strategy, a ceramide-unrelated compound named E16A was established, which potently inhibited the function of CERT in cultured human cells; its official name is (1*S*, 2*R*)-HPCB-5, after its chemical name 4′-(2-hydroxyethanesulfonyl)-4-pentyl-3-(2-(pyridin-2-yl)-(1*S*, 2*R*)-cyclopropyl)-1,1′-biphenyl (Figure 6A) [28]. The compound E25A, in which the *n*-pentyl moiety of E16A is replaced with a cyclopentyl group, was developed as another potent CERT inhibitor (Figure 6A) [28].

Ceramide has just three polar groups, 1-hydroxyl, 3-hydroxyl, and 2-amido groups (Figure 1A). All three of these polar groups form a hydrogen-bonding network with specific amino acid residues in the CERT START domain (Figure 4A). Likewise, polar groups of the ceramide-mimetic, i.e., (1*R*,3*S*)-HPA compounds, and -nonmimetic competitive inhibitors of CERT form hydrogen-bonding networks in the START domain (Figure 4B and Figure 6B) [28,57]. In addition, the pyridyl group of E16A and E25A may form a water molecule-intervening hydrogen-bonding network in the hydrophobic cavity of the START domain (Figure 6B), whereas natural ceramide and HPA-12, which have no polar group at the corresponding position, are incapable of forming such a network (Figure 4) [28]. Van der Waals interactions between the hydrophobic moiety of ceramide and the inner surface of the ceramide-binding pocket in the START domain are also crucial for ligand recognition by CERT [37], and E16A as well as (1*R*,3*S*)-HPA is similarly recognized by the CERT START domain (Figure 4 and Figure 6B) [28,57,63]. Moreover, the three aryl groups and *n*-pentyl groups of E16A (and the cyclopentyl group of E25A) enable the compound to sufficiently occupy the hydrophobic pocket of the START domain (Figure 6B).

The *K*_d_ values of E16A and (1*R*,3*S*)-HPA for the CERT START domain immobilized on an SPR sensor chip are ~60 nM and ~30 nM, respectively (Figure 3 and Figure 6A). In the case of the ceramide-mimetic inhibitor (1*R*,3*S*)-HPA-12, its enantiomer, namely (1*S*,3*R*)-HPA-12, has no affinity (*K*_d_ >10 µM) for the CERT START domain (Figure 3) [28]. In contrast, in the case of the ceramide-nonmimetic inhibitor E16A, its enantiomer (named E16B) has a lower but substantial affinity for the CERT START domain (*K*_d_ = ~1 µM) (Figure 6A) [28]. When estimated by the metabolic labeling method performed in a 10% serum-containing culture medium, the half-maximal inhibitory concentration (*IC*_50_) values of E16A and (1*R*,3*S*)-HPA for CERT in HeLa cells were ~0.2 µM [28]. When the cell-associated level of the two compounds during prolonged (72 h) culture was examined, the level at which E16A plateaued was 3- to 4-fold higher than the level of HPA-12 [28], suggesting that in cell culture E16A is more stable and exerts more prolonged activity than HPA-12.

The establishment of ligand-mimetic, such as (1*R*,3*S*)-HPA-12, and nonmimetic, such as E16A, inhibitors that target CERT has provided a pharmacological tool to discriminate on-target effects from off-target effects (Figure 6C).

## 8. Limonoids Might Inhibit CERT via Nonconventional Mechanisms

The LTP-mediated inter-membrane transfer of a lipid involves at least four biophysical steps [1]: (i) an LTP binds to the donor membrane, (ii) the LTP extracts a lipid from the donor membrane by enclosing the hydrophobic moiety of the lipid in its hydrophobic pocket, (iii) the lipid-holding LTP moves to the acceptor membrane, and (iv) the LTP releases the lipid to the acceptor membrane. Furthermore, after releasing the lipid to the acceptor membrane, some LTPs (e.g., the cholesterol/PtdIns(4)P exchanging-transfer protein OSBP) catalyze the retrograde transport of another specific lipid from the “acceptor membrane” to the “donor membrane”, to exchange different lipid types between different organelles [87,88].

Limonoids are a class of highly oxygenated, modified terpenoids, which are abundant in citrus fruit. Limonoids were initially identified as phytochemicals responsible for the bitterness of citrus fruit but in recent decades have attracted attention for their diverse bioactivities, including anti-oxidative, anti-inflammatory, anti-cancer, and anti-bacterial/viral activities (reviewed in [89,90]). Limonoids have no apparent structural similarity to ceramide (Figure 7A). Hullin-Matsuda et al. found that several limonoids impair the de novo synthesis of sphingomyelin in cells and revealed that limonoids inhibit the CERT-dependent extraction of a long-chain ceramide from the membrane but do not affect the CERT-dependent inter-membrane transfer of the short-chain C_5_-DMB-ceramide [91]. The physicochemical nature of phospholipid membranes affects the efficiency of CERT-dependent extraction of ceramide from the membrane [92]. Thus, it has been proposed that limonoids somehow alter the physicochemical nature of the ER, thereby inhibiting the function of CERT at the step of extracting ceramide from the ER, without occupying the ceramide-binding pocket of CERT [91].

More recently, another model, in which inactive or less active conformations of CERT are stabilized by limonoids, has been proposed. The hyperphosphorylation of the SRM in CERT inactivates the functioning of CERT, possibly via a conformational alteration to induce mutual interference between the PH and START domains (Figure 2) [36]. Co-crystallography of an isolated PH domain in complex with an isolated START domain suggested that a specific region of the PH domain interacts with a specific region of START [94]. A recent molecular dynamics study has predicted that several limonoids may stabilize the interaction between the PH and START domains of CERT by intercalating into the interface between PH and START domains in the putative inactive conformation of CERT (Figure 7B) [93]. Unfortunately, as it employed the (1*R*,3*R*)-configuration of HPA-12, the previous simulation study suggested that the binding energy of HPA-12 to the CERT START domain was less than that of several limonoids [93]; therefore, a re-run of the simulation would be desirable in the future, to estimate the binding energy of the potent inhibitor (1*R*,3*S*)-HPA-12.

Natural, food-derived small chemicals may provide a relatively safer compound, suitable for use as a CERT inhibitor in humans. However, considering that limonoids exhibit many biological activities [89,90], one should pay attention to any off-target effects that may occur when using limonoids as a CERT inhibitor.

## 9. Possible Applications of CERT Inhibitors

CERT plays a key role in the ceramide-sphingomyelin metabolic axis (reviewed in [11]). As more evidence has accumulated that ceramide and sphingomyelin participate in various physiological and pathophysiological events [15,16,23,24], more attention has been paid to CERT. Accordingly, it has been clarified that CERT and CERT_L_ (also known as GPBPΔ26 and GPBP, respectively) participate in various events and functions in mammalian cells, including polyploid cancer cell death [95], EGF receptor signaling [96], lipotoxicity and glucolipotoxicity in islet β-cells [97,98], muscle insulin signaling [99], myofibril formation [100], integrity of glomerular basement membrane [101], cytotoxic autophagy [102], mitochondrial maintenance [103], and senescence [104].

Interest in CERT inhibitors has also been increasing, although as far as we know CERT inhibitors have yet to be used clinically. One of the most desirable applications of CERT inhibitors may be their therapeutic use in cancer, because the upregulation of intracellular ceramide levels by the blockage of the ceramide-to-sphingomyelin conversion through the use of CERT inhibitors would facilitate the efficacy of existing anti-cancer drugs, as previously proposed [95]. For further possible applications of CERT inhibitors in the treatment of cancer, see the recent review by Chung et al. [88].

Another important application may be the therapeutic or prophylactic use of CERT inhibitors to treat disorders caused by abnormally activated CERT. Rapid advances in human clinical genetics studies have revealed that various specific missense mutations in the human *CERT1* gene are associated with hereditary intellectual disabilities and mental development disorders with an autosomal dominant inheritance pattern [44,45,46,47,48], and some of these mutations were recently shown to compromise the functional repression of CERT [48,49]. Thus, a CERT inhibitor could be a rational tool to reduce abnormally activated CERT to a normal level. This could in turn improve the pathology of the *CERT1* mutation-caused disease without shutting off the physiologically indispensable function of CERT, although more studies will be necessary to identify a clinically usable CERT inhibitor.

As depicted in Figure 2, *CERT1* encodes at least two CERT-related proteins, CERT/GPBPΔ26 and CERT_L_/GPBP, which are transcribed from different splicing transcripts. Revert et al. showed that CERT_L_/GPBP, but not CERT/GPBPΔ26, is exported from cells via a nonconventional secretion pathway [52]. Intriguingly, CERT_L_/GPBP was reported to be capable of binding various extracellular proteins including type IV collagen [50], serum amyloid P-component [105], the complement factor C1q [106], and the amyloid precursor protein and amyloid-β [65]. Thus, CERT inhibitors may also be a useful tool to investigate the physiological and pathophysiological significance of interactions between CERT_L_/GPBP and extracellular proteins, although it remains unclear whether the function of extracellular CERT_L_/GPBP is relevant to its ceramide-binding activity. Crivelli et al. recently revealed that genetically modified enhancement of CERT_L_ expression in the brain in a familial Alzheimer’s disease model in mice reduces the amyloid-β level, accompanied by a decreasing ceramide level concomitant with increasing sphingomyelin level in the brain; they also found that administration of HPA-12 to the model mice for 4 weeks exacerbated the ceramide level and amyloid-β pathology [65]. Thus, long-term administration of a CERT inhibitor might aggravate Alzheimer’s disease. If ceramide-binding activity is also crucial to the function of extracellular CERT_L_/GPBP, then membrane-impermeable compounds, rather than membrane-permeable compounds, will be more appropriate to selectively modulate their extracellular function(s) without inhibiting their intracellular function(s).

The pharmacological or genetic inhibition of CERT negatively affects the proliferation of several types of intracellular pathogens, including hepatitis C virus [107,108,109], rubella virus [110], and the obligate intracellular bacterium Chlamydia [9,111,112]. Hence, CERT inhibitors might be applicable as anti-infectious disease agents, although pathogen genome-encoded factors, rather than human genome-encoded factors, may be more suitable molecular targets of anti-infectious disease drugs in terms of the risk of side effects. On this point, it is noteworthy that a stereoisomer of HPA-12, which has no activity to inhibit CERT or human SMS, compromises the proliferation of *Chlamydia trachomatis*, presumably by inhibiting the activity of an enzyme tentatively named a Chlamydia sphingomyelin synthase (K. Kumagai, S. Sakai, M. Kataoka, M. Ueno, S. Kobayashi, and K. Hanada, manuscript in preparation).

We would like to briefly discuss feasible strategies to reduce various possible off-target effects of CERT inhibitors. First, compared with ceramide-mimetic inhibitors, nonmimetic inhibitors should be less likely to affect other proteins that share natural ceramides as their ligands. Second, the design of compounds to minimize undesirable metabolic alterations is beneficial to reduce off-target effects: for instance, with ceramide-mimetic CERT inhibitors, their terminal OH groups might be metabolically adducted with polar groups in cells (as natural ceramides are converted to sphingomyelin and glucosylceramide), while their *N*-acyl groups might be hydrolyzed by ceramidases and other amidases; these metabolic derivatives might then also act as sphingolipid-like bioactive molecules. By contrast, the ceramide-nonmimetic inhibitor E16A is anticipated to be nonsusceptible to these ceramide metabolic reactions, which may minimize the adverse effects triggered by metabolites of the inhibitor. Third, the coordination of drug design and delivery systems may help to reduce off-target effects. For example, if an inhibitor is not intended to affect the function of the brain, blood–brain barrier-impermeable derivatives of the inhibitor should be designed. Conversely, if an inhibitor is intended to be delivered to the brain, advanced technologies for the delivery of a drug from the peripheral blood to the brain would be useful.

## 10. Concluding Remarks and Perspectives

In the past two decades, numerous LTPs have been revealed to be central to intracellular lipid trafficking and to play key roles in the formation and regulation of the lipidome in the cell. Accordingly, LTPs have been or will be attracting attention as a novel type of molecular target for medicinal products. As summarized in this short review, several CERT inhibitors have been developed via various research routes, including serendipitous discovery [54], and computer-assisted drug design [28]. Although de novo drug development has been the predominant route used to identify compounds that are effective for specific diseases, the alternative route via “drug repositioning” can be less costly than de novo drug development. This is because a drug’s toxicity and pharmacokinetic properties are already understood; additionally, drug-repositioning movements have been boosted globally by the recent need to find anti-COVID-19 drugs as soon as possible [113]. Furthermore, natural food ingredients are often safer than nonnatural compounds, because the safety of natural foods has been proved practically throughout the long history of humans’ daily diets. Thus, discovering new LTP inhibitors from among existing approved drugs and natural food ingredients is encouraged, in parallel with the de novo development of novel compounds.

All bioactive chemicals inevitably have side effects. Particularly for agonists/antagonists that are structurally mimetic to natural ligands on the molecular target, they may exhibit promiscuous interactions with various targets sharing the same natural ligand. Therefore, the development of at least two types of structurally unrelated or at least very different inhibitors to one LTP class should be encouraged, because they provide a good pharmacological tool to assess the specific impacts of LTP inhibition on the physiology and pathophysiology of living organisms, compared with the examination with only one type, as conceptually depicted in Figure 6C.

It might be difficult to find a small chemical antagonist that specifically targets one member among an LTP family having mutually resembling ligand-binding domains, for instance, the OSBP-related protein (ORP) family [114]. Indeed, a class of sterol-mimetic and nonmimetic compounds that exhibit anti-cancer and anti-enteroviral activity target two sterol transfer proteins, OSBP and ORP4 (also known as OSBP2) [115,116]. Nevertheless, even when members of a protein family share structural similarities, various selective or specific agonists/antagonists to one member of the family can be developed, for example (e.g., protein kinases [117,118], G protein-coupled prostanoid receptors [119], and the nuclear retinoid receptors RAR/RXR [120]). It is hoped that this successful pharmaceutical history is also adaptable to human LTPs.

## Figures and Tables

**Figure 1 ijms-23-02098-f001:**
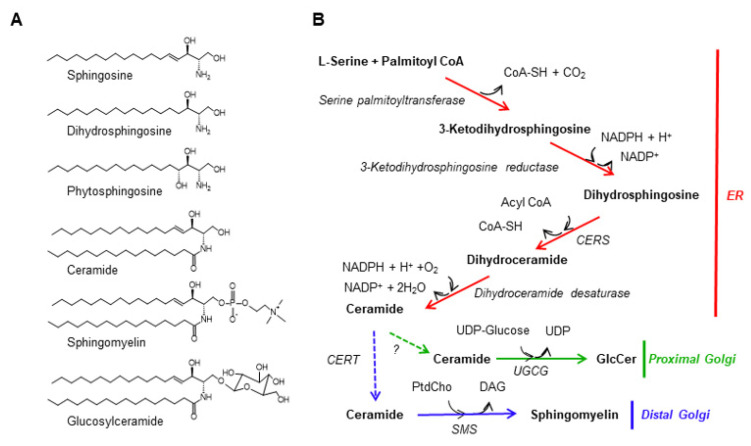
The de novo synthetic pathway for sphingomyelin. (**A**) Structure of mammalian sphingolipids. The chemical structures of three major sphingoid bases and several complex sphingolipids in mammalian cells are shown. Ceramide, sphingomyelin, and GlcCer are depicted as the *N*-palmitoyl sphingosine type, although there are various combinations of sphingoid bases and *N*-acyl chains in cells. (**B**) The de novo synthetic pathways for sphingomyelin and GlcCer in mammalian cells are shown. Solid arrows represent enzymatic reactions while dotted arrows represent inter-organelle transport processes. PtdCho, phosphatidylcholine.

**Figure 2 ijms-23-02098-f002:**
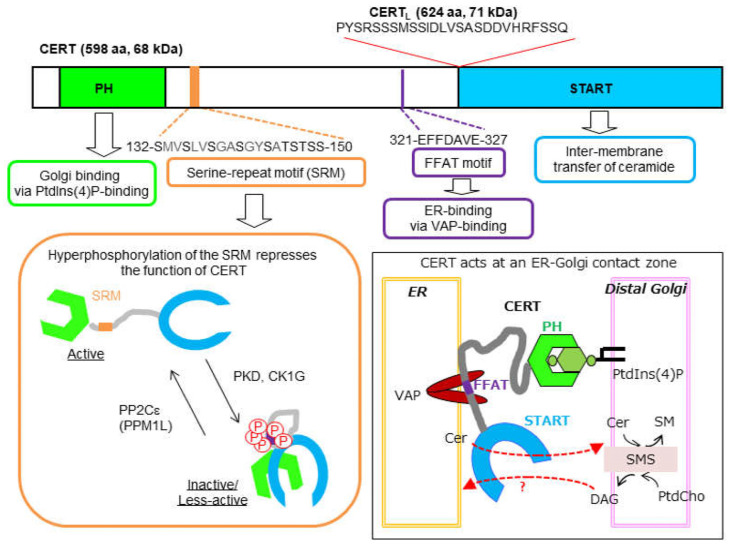
Functional domains and motifs of CERT. The functions of the domains and motifs in human CERT are shown. The numbering represents the amino acid number, starting from the initial methionine. Amino acid sequences of the SRM and the core seven amino acids in the FFAT motif of CERT are indicated. CERT_L_, which is the product of a splicing variant transcript of *CERT1*, has an extra sequence of 26 amino acids in front of the START domain. Hyperphosphorylation of the SRM inactivates CERT, possibly via a conformational change, to induce mutual interference between the PH and START domains [36]. Cer, ceramide; SM, sphingomyelin; PtdCho, phosphatidylcholine; PKD, protein kinase D; CK1G, casein kinase 1γ isoform; PP2Cε (or PPM1L), protein phosphatase 2Cε (or protein phosphatase Mg^2+^- or Mn^2+^-dependent 1L). Inset, how CERT executes the ER-to-Golgi trafficking of ceramide at an ER–Golgi contact zone is schematically illustrated. It remains unclear whether CERT mediates Golgi-to-ER trafficking of DAG in exchange for ER-to-Golgi trafficking of ceramide in living cells, although CERT has been demonstrated to catalyze the inter-membrane transfer of long-chain DAG [27] in a cell-free system. It should be noted that for simplicity, CERT is illustrated as a monomer in the schematic model, although CERT most likely forms homo-oligomers in cells [40].

**Figure 3 ijms-23-02098-f003:**
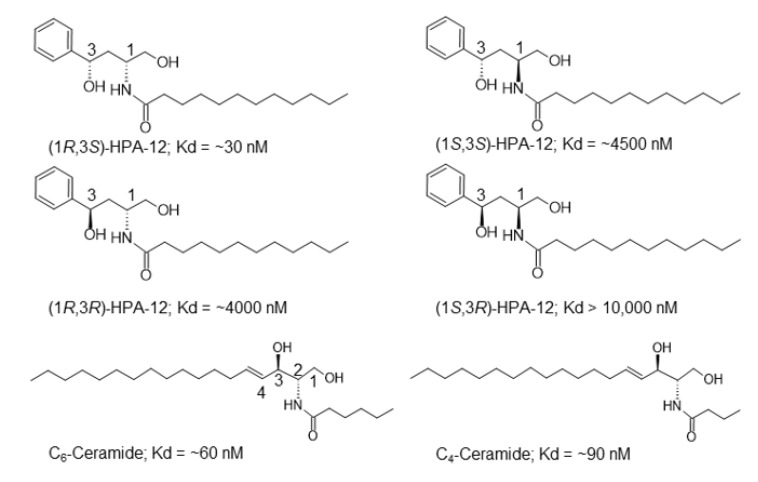
(1*R*,3*S*)-HPA-12, a ceramide-mimetic inhibitor of CERT, and its relevant compounds. The structure of (1*R*,3*S*)-HPA-12, its stereoisomers, and short-chain ceramides (C_4_-ceramide and C_6_-ceramide) are shown with their *K*_d_ values for the CERT START domain, which were determined by SPR analysis [28].

**Figure 4 ijms-23-02098-f004:**
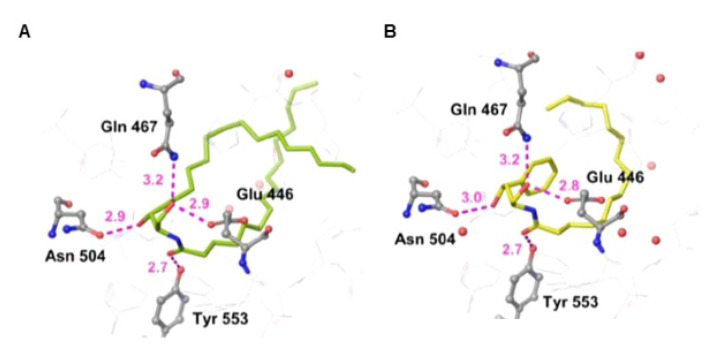
Crystallography of the CERT START domain in complex with natural ceramide and a ceramide-mimetic inhibitor. Co-crystallography images of the CERT START domain in complex with C_16_-ceramide (**A**) and (1*R*,3*S*)-HPA-15 (**B**), which were originally presented in [28], under the Creative Commons (CC) BY conditions, are shown. Magenta dashed lines indicate hydrogen bonds (distances are indicated in angstroms).

**Figure 5 ijms-23-02098-f005:**
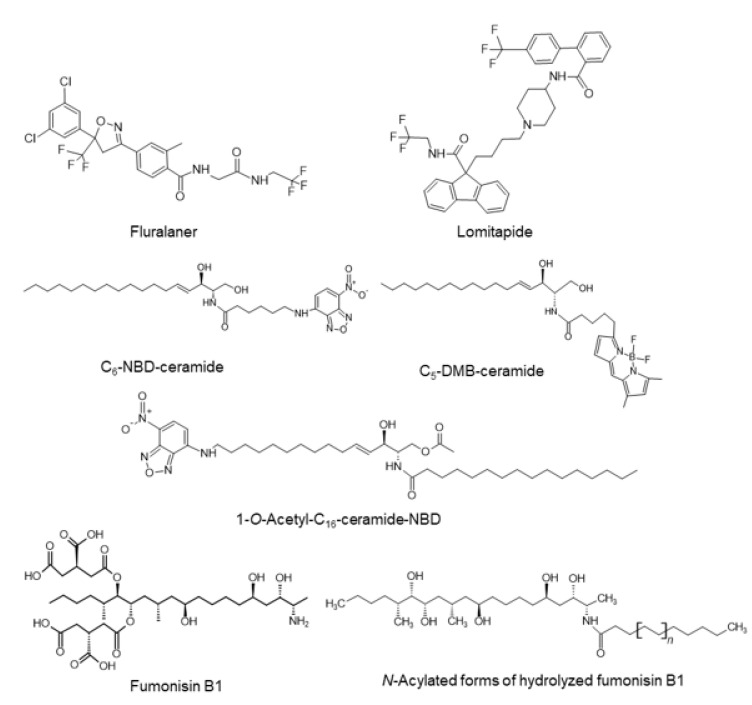
Various ceramide-mimetic compounds can serve as CERT inhibitors. The chemical structures of ceramide-mimetic compounds that are described in this article are shown. It should be noted that whether fluralaner and lomitapide can be categorized as ceramide-mimetic compounds is unclear because they share only a minimum structural resemblance (i.e., hydrophobic amido groups linked to other hydrophobic moieties). Various fluorescent analogs of ceramide are recognized by CERT, meaning that they compete with endogenous natural ceramide in cells. The sphingoid base-like compound fumonisin B1 acts as a potent inhibitor of the ceramide synthases. When fumonisin B1 is metabolized to an *N*-acylated form, the resultant ceramide-like compound may act as an inhibitor of CERT.

**Figure 6 ijms-23-02098-f006:**
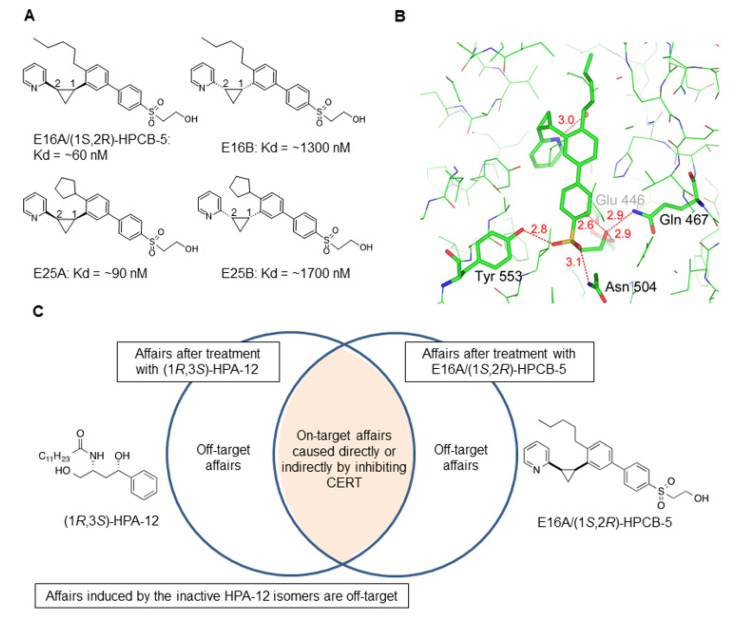
E16A/(1*S*, 2*R*)-HPCB-5, a ceramide-nonmimetic inhibitor of CERT, and its relatives. (**A**) The structures of E16A/(1*S*, 2*R*)-HPCB-5 and its relatives are shown with their *K*_d_ values for the CERT START domain, which were determined by SPR analysis [28]. (**B**) A co-crystallography image for the CERT START domain in complex with E16A/(1*S*, 2*R*)-HPCB-5, which was originally presented in [28], under CC BY conditions, is shown. Red sphere, the oxygen atom of the water molecule; red dashed lines, hydrogen bonds (distances are indicated in angstroms). (**C**) A schematic diagram representing the notion that the establishment of the ligand-mimetic and -nonmimetic inhibitors that target CERT has provided a pharmacological tool to discriminate on-target effects from off-target effects.

**Figure 7 ijms-23-02098-f007:**
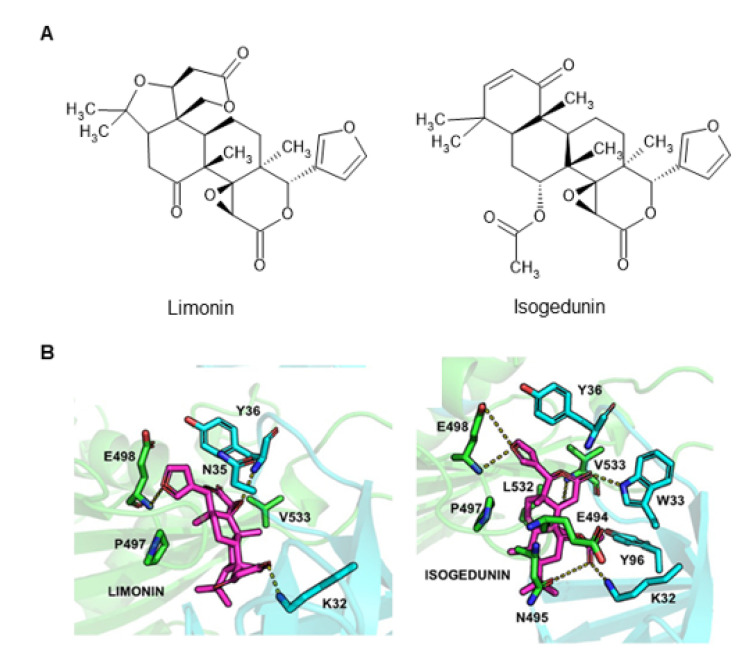
Limonoids as CERT inhibitors. (**A**) The structures of two limonoids are shown. (**B**) Limonoids might stabilize a putative inactive conformation of CERT by intercalating into the interface between the PH and START domains. Computer simulation models for limonin *(left panel*) and isogedunin *(right panel*), which were originally presented in [93], under CC BY conditions, are shown. In each panel, magenta, blue, and green represent the limonoid, CERT PH domain, and CERT START domain, respectively. Yellow dashed lines represent hydrogen bonds.

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
