# Peer review of "Natural Ligand-Mimetic and Nonmimetic Inhibitors of the Ceramide Transport Protein CERT"

_ijms, 2022, doi:10.3390/ijms23042098_

Round 1

Reviewer 1 Report

Hanada et al. covered nice review article extensively on the ceramide transport protein CERT and its potential inhibitors and biological applications. However, I have observed a few significant drawbacks in this review manuscript. The followings are.

  1. Line 501-509, the authors made statements that “Several types of CERT inhibitors have been developed via various research routes,-including from serendipitous discovery to in silico-based approaches. How and what based the authors confirming this statement by just in silico strategies. I have not seen any solid reference cited to claim their mechanism of inhibition.
  2. Line 509-511, “Finding new LTP inhibitors from existing approved medicines (or “drug repositioning”) and natural food ingredients may be more cost-effective than de novo development of novel compounds" no clarity in the above statement.
  3. “Limonoids inhibit CERT via a non-conventional mechanism," The author stated that Limonoids have no apparent structural similarity to ceramide. The authors claim two statements in the same article without proper evidence inhibits CERT's non-conventional targets.
  4. The manuscript needs substantial corrections and English language in the entire manuscript. I suggest that the authors go for native English speakers and publish broad scientific readers.

Author Response

Response to Reviewer 1 Comments

Hanada et al. covered nice review article extensively on the ceramide transport protein CERT and its potential inhibitors and biological applications. However, I have observed a few significant drawbacks in this review manuscript. The followings are.

Point 1: Line 501-509, the authors made statements that “Several types of CERT inhibitors have been developed via various research routes,-including from serendipitous discovery to in silico-based approaches. How and what based the authors confirming this statement by just in silico strategies. I have not seen any solid reference cited to claim their mechanism of inhibition.

Response 1: The development of HPA-12 is an example of serendipitous discovery. The structure-based drug design, which was used for the development of E16A, employed an “in silico-based approach” in the first screening step. To avoid misunderstanding, we changed the term “in silico-based approach” to “computer-assisted drug design”, and revised the sentence as follows: “As summarized in this short review, several CERT inhibitors have been developed via various research routes, including serendipitous discovery (Yasuda et al., 2001), and computer-assisted drug design (Nakao et al., 2019).” (Line 448-449)

Point 2: Line 509-511, “Finding new LTP inhibitors from existing approved medicines (or “drug repositioning”) and natural food ingredients may be more cost-effective than de novo development of novel compounds" no clarity in the above statement.

Response 2: Thanks to the reviewer’s comment, we recognized the necessity of an explanation on the cost-effectiveness of drug-repositioning and natural food ingredients, and revised the part as follows: briefly explained why they may be cost-effective as below.

“Although de novo drug development has been the predominant route to identify compounds effective for specific diseases, the alternative route via “drug repositioning” can be less costly than de novo drug development, because the drugs’ toxicity and pharmacokinetic properties are already understood, and, therefore, the drug repositioning movements have been globally boosted by the emergent aim to find anti-COVID-19 drugs as soon as possible {Bayoumy, 2020 #579}. In addition, natural food ingredients are often safer than non-natural compounds, because the safety of natural foods has been practically proved during the long history of daily diets of humans. Thus, finding new LTP inhibitors from existing approved drugs and natural food ingredients will also be encouraged, in parallel with the de novo development of novel compounds.” (Line 449-457)

  In addition, at the first appearance of the term “drug repositioning”, a short explanation of the term was added as follows (underlined parts were added in the revised version): “drug repositioning (also referred to as drug repurposing and drug reprofiling)”, which is the scientific strategy of revisiting existing drugs for new therapeutic purposes”. (Line 251-252).

Point 3. “Limonoids inhibit CERT via a non-conventional mechanism," The author stated that Limonoids have no apparent structural similarity to ceramide. The authors claim two statements in the same article without proper evidence inhibits CERT's non-conventional targets.

Response 3: Thank you for the careful comment. Because the two papers cited have proposed mutually different mechanisms, and it remains unclear which mechanism is appropriate, we corrected the subheading to "Limonoids might inhibit CERT via non-conventional mechanisms”. (Line 333)

In addition, to discriminate the proposed models from the conventional “lipid-binding pocket occupying” mechanism, two original sentences were changed as below (underlined parts were added in the revised version):

“it has been proposed that limonoids somehow alter the physicochemical nature of the ER, thereby inhibiting the function of CERT at the step of extracting ceramide from the ER without occupying the ceramide-binding pocket of CERT {Hullin-Matsuda, 2012 #526}.” (line 353)

 “More recently, another model, in which inactive or less active conformations of CERT are stabilized by limonoids, has been proposed.” (Line 355-356)

Point 4. The manuscript needs substantial corrections and English language in the entire manuscript. I suggest that the authors go for native English speakers and publish broad scientific readers.

Response 4: The original manuscript has been already proofread by a native English speaker. However, we amended several parts as specified in the responses to other comments of the reviewer. In addition, other parts listed below were corrected.

Line 13, and Line 446-467: “molecular medicine target” was changed to “molecular target for medicinal products”.

Line 444: “not only CERT but also numerous LTPs” was simplified to “numerous LTPs“.

Line 465: “One may feel the difficulty find a small compound---” was changed to “It might be difficult to find a small chemical antagonist---”.

Reviewer 2 Report

This is is well constructed review describing the recent advances in development and usage of CERT inhibitors and suitable for publication in IJMS.

Comments: 

Sphingolipids, particularly sphingomyelin is one of the essential components of mammalian membranes. Dysregulation of sphingolipid pathways can have unintended side effect on membrane fluidity. Although CERT inhibitors can be used as potential therapeutic agents, the ubiquitous distribution of CERT and sphingolipids across various tissues could prevent the use of these inhibitors in clinics. These side effects are likely to be independent of the chemical nature of the inhibitors ( ceramide analog or structurally distinct classes). It would be ideal  if the authors could include some potential ways to mitigate the side effect of these inhibitors in Section 9: "Possible applications of CERT inhibitors". They could possibly describes the use of targeted delivery approaches, that has been tested other potential drug candidates.  

Author Response

Point 1: It would be ideal if the authors could include some potential ways to mitigate the side effect of these inhibitors in Section 9: "Possible applications of CERT inhibitors". They could possibly describes the use of targeted delivery approaches, that has been tested other potential drug candidates.

Response 1: Thank you very much for the constructive comment. Because extended explanation on this point is presumably beyond the scope of the short review, we added one paragraph described below to section 9. We hope the brief discussion would meet the reviewer’s request.

“We would like to briefly discuss feasible strategies to reduce various possible off-targeting effects of CERT inhibitors. First, compared with ceramide-mimetic inhibitors, nonmimetic inhibitors should have less probability to affect the other proteins sharing the natural ceramide as their ligands. Secondly, compound design to minimize undesired metabolic alterations is presumably beneficial to reduce off-targeting effects: For instance, in ceramide-mimetic CERT inhibitors, their terminal OH groups might be adducted by polar groups as natural ceramide while their N-acyl groups might be hydrolyzed by ceramidases and other amidases, and then, these metabolic derivatives might also act as sphingolipid-like bioactive molecules. By contrast, the ceramide-nonmimetic inhibitor E16A is expected to be non-susceptible to these ceramide metabolic reactions, which may minimize the adverse effects triggered by metabolites of the inhibitor. Thirdly, coordination of drug design and delivery systems may reduce off-targeting effects. For example, if an inhibitor is undesired to affect the function of the brain, blood-brain barrier-impermeable derivatives of the inhibitor should be designed. By contrast, if an inhibitor is wanted to deliver into the brain, advanced technologies for the delivery of a drug from the peripheral to the brain might be usable.” (Line 427-441)

Reviewer 3 Report

The review from Hanada and co-authors is very well written and is very interesting. It highlights the recent progress made in the development of CERT inhibitors and their possible therapeutic application. The review is acceptable in the present form.

Author Response

Point 1: The review from Hanada and co-authors is very well written and is very interesting. It highlights the recent progress made in the development of CERT inhibitors and their possible therapeutic application. The review is acceptable in the present form.

Response 1: Thank you very much for your favorite evaluation of our manuscript.

Round 2

Reviewer 1 Report

The authors addressed all the queries. No further comments